# Severe COVID-19—A Review of Suggested Mechanisms Based on the Role of Extracellular Matrix Stiffness

**DOI:** 10.3390/ijms24021187

**Published:** 2023-01-07

**Authors:** Garry Kerch

**Affiliations:** Faculty of Materials Science and Applied Chemistry, Riga Technical University, 1048 Riga, Latvia; garrykerch@inbox.lv

**Keywords:** COVID-19, cardiovascular diseases, stiffness, mechanosensitive platelets, thrombotic complications

## Abstract

The severity of COVID-19 commonly depends on age-related tissue stiffness. The aim was to review publications that explain the effect of microenvironmental extracellular matrix stiffness on cellular processes. Platelets and endothelial cells are mechanosensitive. Increased tissue stiffness can trigger cytokine storm with the upregulated expression of pro-inflammatory cytokines, such as tumor necrosis factor alpha and interleukin IL-6, and tissue integrity disruption, leading to enhanced virus entry and disease severity. Increased tissue stiffness in critically ill COVID-19 patients triggers platelet activation and initiates plague formation and thrombosis development. Cholesterol content in cell membrane increases with aging and further enhances tissue stiffness. Membrane cholesterol depletion decreases virus entry to host cells. Membrane cholesterol lowering drugs, such as statins or novel chitosan derivatives, have to be further developed for application in COVID-19 treatment. Statins are also known to decrease arterial stiffness mitigating cardiovascular diseases. Sulfated chitosan derivatives can be further developed for potential use in future as anticoagulants in prevention of severe COVID-19. Anti-TNF-α therapies as well as destiffening therapies have been suggested to combat severe COVID-19. The inhibition of the nuclear factor kappa-light-chain-enhancer of activated B cells pathway must be considered as a therapeutic target in the treatment of severe COVID-19 patients. The activation of mechanosensitive platelets by higher matrix stiffness increases their adhesion and the risk of thrombus formation, thus enhancing the severity of COVID-19.

## 1. Introduction

COVID-19 has become a major global concern in the last two years [1]. The high expression of intercellular adhesion molecule-1 (ICAM-1) by endothelial cells in COVID-19 patients was reported [2]. It has been found that severity of the disease increases with aging and demonstrated [3] that increased adhesion molecule ICAM-1 expression can increase virus entry in COVID-19 due to increased tissue permeability, and at the same time, increased adhesion molecule ICAM-1 expression can increase the risk of cardiovascular diseases due to tight junction integrity disruption and the increased transmigration of neutrophils [2], Figure 1. Vascular cell adhesion molecule 1 (VCAM-1) and ICAM-1 adhesion molecule have been recently reported as circulating markers of arterial stiffness [4]. Post-COVID-19 complications can cause negative symptoms that last weeks or months after the infection have gone. Potential long-lasting effects of 6 months of SARS-CoV-2 infection on markers of arterial stiffness VCAM-1 and ICAM-1 among young adults demonstrated negative long-term cardiovascular health problems following COVID-19 infection.

Arterial stiffness determined from carotid-femoral pulse wave velocity for mild, moderate, and severe COVID-19 patients has been found to be essentially higher for severe and moderate COVID-19 patients compared to mild COVID-19 patients [5].

The elevated plasma levels of adhesion molecules, such as ICAM-1 and VCAM-1, vascular adhesion protein-1, and vascular endothelial growth factor, has been reported for severe and moderate COVID-19 patients [2,6].

The increased ICAM-1 expression may take place as a result ofthe increase of pro-inflammatory cytokines, such as TNF-α and/or cytoskeletal reorganization. Selenium can inhibit tumor necrosis factor alpha (TNF-α) induced ICAM-1, VCAM-1, and endothelial-leukocyte adhesion molecule-1 (E-selectin) expression [7]. It has been also reported that resveratrol prevents TNF-α-induced VCAM-1 and ICAM-1 upregulation in endothelial progenitor cells via the reduction of the nuclear factor kappa-light-chain-enhancer of activated B cells (NF-κB) activation and translocation from the cytoplasm into the nucleus [8]. During the aging process, the release of tightly bound water molecules in extracellular matrix (ECM) [9,10,11], dehydration, glycation, and crosslinking by advanced glycation end products increases tissue stiffness and related cytoskeletal reorganization and pro-inflammatory cytokines production [9,10,11]. The content of water that is bound to collagen decreases with aging and collagen cross-links multiply with age increasing tissue stiffness [12]. A strong correlation between adhesion strength of cells and hydrated water content of polymer substrates has been suggested [13]. Water loosely bound at the collagen and water tightly bound within collagen result in different extracellular matrix properties [14].

Platelet hyperactivity and thrombotic complications in severe COVID-19 patients with comorbidities such as hypertension, cardiovascular disease, and obesity, but not in mild COVID-19 patients, was reported. Thrombotic complications are a well-recognized hallmark of COVID-19. The role of the thrombotic complications in COVID-19 is not completely understood at present. Manne and coauthors consider that “pathological drivers of thrombosis in COVID-19 remain unclear” [15].

Platelets play important role also in processes in lungs. It has been reported that higher numbers of platelets are associated with higher rates of venous thromboembolism, but higher mean platelet volume is associated with higher rates of arterial thrombotic events [16].

The review will focus on the influence of micro-environmental stiffness on various mechanisms of cellular processes involved in COVID-19 that have been suggested by different authors. Essential improvement in mechanisms understanding can be achieved if increased arterial stiffness in critically ill COVID-19 patients is taken into account. This review will cover only publications devoted to the various aspects of extracelllular matrix (ECM) stiffness interrelated with oxidative stress and the inflammation that has been observed in the COVID-19 disease with the aim to provide a rational basis for the future development of novel therapies. Publications related to direct and indirect destiffening strategies and destiffening agents are of great interest. The effect of mechanical forces and extracellular matrix stiffness on biological cellular processes was the criterium for published data inclusion in this review.

## 2. Possible Causes of COVID-19 Severity

### 2.1. Dehydration and Tissue Stiffness

The effect of the combination of surface stiffness with surface hydration on the cellular processes has been reviewed [10] and it was demonstrated that endothelial cells are mechanosensitive, and the increase in arterial stiffness with aging leads to compromised cell–cell junction integrity with leukocyte and neutrophils transmigration into the intima. Greater cholesterol uptake was also reported. All these processes result in atherosclerotic plaque formation and thrombosis development [9,10,11].

Dehydration also results in enhanced tissue stiffness and triggers NF-κB activation [11]. Hydration at biointerfaces prevents platelet adhesion and thrombus formation [3].

Patients with severe COVID-19 admitted to the intensive care unit (ICU) are at especially high thrombotic risk [17]. It was also observed that atherosclerosis, diabetes, and obesity are associated with the stiffening of blood vessel walls and the recruitment of inflammatory leukocytes out of the bloodstream into tissues. It has been demonstrated that the decrease of water molecules binding energy with biological macromolecules leads to the release of tightly bound water and the increase of ECM stiffness with the disruption of tissue integrity [3].

### 2.2. Platelets Are Mechanosensitive

Platelets are mechanosensitive, and increasing substrate stiffness results in the increased platelet adhesion and spreading [18,19]. Adhesion on stiffer substrates can be related with higher levels of platelet activation. Substrate stiffness-dependent platelet adhesion and activation are related with Rac1 and actomyosin activity [20,21]. The inhibition of actin polymerization reduces stiffness and decreases invasiveness of cells. Inflammatory cytokines TNF-α and interleukins (ILs) (IL-6 and IL-8) increase platelet reactivity and the activation of leukocytes leading to a pro-thrombotic state [22]. Tissue integrity plays important role in COVID-19 [3]. Increase in tissue permeability as a result of ECM stiffening increases the incidence and severity of COVID-19. Thrombus formation as a result of platelet activation can complicate the treatment of COVID-19. Catheter-related thrombosis was reported in patients with severe COVID-19 admitted to ICU [23]. So, the development of antibacterial, antiviral, and anticoagulant coatings for catheters used in critically ill COVID-19 patients is urgently needed. Surfaces coated with heparin reduce platelet adhesion and activation. Low molecular weight heparin is recommended for hospitalized patients with COVID-19 due to the concerns of thrombus formation that have been discussed in the recent Consensus Statement [24]. Heparin-like sulfated chitosan also has potential to be used as heparin alternative due to its ability to decrease platelet adhesion [19], Figure 1.

Increased arterial stiffness and the increased content of membrane cholesterol was observed in age-related diseases and in age-dependent COVID-19. The increase of membrane cholesterol content increases arterial stiffness. Increased arterial stiffness can be associated with increased platelet adhesion, platelet activation and thrombus formation. The increase of membrane cholesterol content enhances the probability of virus entry and severity of COVID-19. Heparin and chitosan sulfates have the potential to mitigate thrombus formation. Cytokine storm induced by increased vascular wall stiffness inputs both into tight junction integrity disruption and into platelets activation. The negative effect of increased tissue stiffness can be reversed by de-stiffening technologies [3].

Various molecular mechanisms have been suggested to describe infection process. For example, Iwasaki and coworkers consider that the S spike protein of SARS-CoV-2 binds with angiotensin-converting enzyme 2 (ACE2) as a receptor and then enters into host cells. The hyper-proinflammatory cytokine storm, including interleukin-6, nuclear factor kappa B (NF-κB), and TNF-α released from SARS-CoV-2-infected macrophages and monocytes, results in multi-organ injury/failure and the development of severe COVID-19 [25]. ACE2 and IL6 inhibitors play an important role in COVID-19. The high expression of ACE2 in the human lung leads to the release of IL6 by suppressing cellular immunity [26]. Mechanosensitivity plays important role in cellular processes. Nasrullah Faisal et al. demonstrated that the attachment of ACE2 with SARS-CoV-2 and SARS-CoV viruses result in an increased stiffness of ACE2. The change in the stiffness of ACE2 was six times higher for SARS-CoV-2 than for SARS-CoV [27].

### 2.3. TNF-α Activation Decreases Tissue Integrity and Increases Stiffness

Correlation between platelet activation and arterial stiffness was reported by Yamasaki and coauthors [28], although the exact mechanism has not been suggested. Increased arterial stiffness is the result of arteriosclerosis. Carotid-femoral pulse wave velocity was used to measure arterial stiffness. The correlation of pulse wave velocity with C-reactive protein, IL-1β, IL-6, TNF-α, and von Willebrand factor (vWF) was reported [29]. Inflammation is closely related with tissue stiffness. vWF plays a major role in blood coagulation, which means that arterial stiffness can play a major role in blood coagulation, platelet activation and adhesion, and thrombus formation, as seen in Figure 2.

The hyperexpression of Factor VIII in COVID-19 subjects was observed. It was suggested that the thrombogenicity of SARS-CoV-2 infection might be linked to widespread endothelial damage and Factor VIII [30,31,32,33]. Complex interaction among endothelial cells and circulating cells have been described in the suggested process of thrombus formation. The release of the von Willebrand factor complexed in blood with factor VIII from endothelial cells exposed to blood laminar flow is stimulated by the arterial wall under mechanical stress. Mechanical shear stress increases the synthesis of factor VIII [34,35].

Dehydration and glycation reactions during aging process leads to the increase of ECM density and stiffness. Changes in ECM stiffness result in cytoskeleton reorganization and concomitant transcription factor NF-κB activation. TNF-α is the inducer of the nuclear transcription factor NF-κB. TNF-α activation decreases tight junction integrity and leads to leukocytes extravasation and to related cardiovascular diseases. At the same time, tight junction integrity disruption leads to the increased virus entry and development of severe COVID-19 disease [3].

### 2.4. NF-κB Has Links to Thrombotic Processes and Inflammation

The role of transcription factor NF-κB was demonstrated in Figure 3. Increased ECM stiffness and related cytoskeletal reorganization activate transcription factor NF-κB [36] and TNF-α, resulting in tight junction integrity disruption and increased adhesion molecule ICAM-1 expression. Tight junction integrity disruption leads to increased transmigration of neutrophils, platelet and leukocyte adhesion, and transendothelial migration and cardiovascular diseases and increased ICAM-1 expression facilitates virus entry and severe COVID-19 development.

The reorganization of the cytoskeleton induced by increased aging and increased density of ECM due to crosslinking, dehydration, and glycation reactions leads to the activation of transcription factor NF-κB. Both inflammation and thrombotic complications depend on the activation of transcription factor NF-κB. The expression of adhesion molecules leads to the binding and transmigration of leukocytes, but virus entry increases vascular wall stiffness and initiates in critical points with high stiffness platelet adhesion and thrombus formation [3].

### 2.5. RHOA Activity Increases with Increase of Stiffness

It has been reported that Ras homolog family member A (RhoA) activity increases with the increase of stiffness, but simvastatin treatment decreases active RhoA levels. Simvastatin decreases endothelium permeability is due to the decrease of ECM stiffness [37]. The activation of RhoA GTPase and its downstream effector, Rho kinase (ROCK), contributes to coagulation in pulmonary endothelial cells. Rho kinase inhibitors have been suggested to combat COVID-19 [38]. Virus penetration into damaged blood vessels increases dramatically local rigidity and trigger thrombosis development. Young’s modulus value of different virus particles is about 45–1000 Mpa [39], but cell stiffness varies from 0.01 to 1000 kPa [40]. So, virus penetration results in the formation of local area in vascular wall with high rigidity where the activation of mechanosensitive platelets leads to thrombus growth. The localized increase of substrate stiffness, for example, as a result of virus penetration or dehydration and glycation processes, can trigger thrombotic complications [3].

Substrate stiffness leads to Rho signaling. Rho signaling initiates NF-Kb signaling and increases EC permeability. NF-Kb signaling results in the expression of adhesion molecules and cytokines and leads to ECM synthesis, increasing substrate stiffness. Rho signaling increases cellular stiffness that leads to ICAM-1 clustering and leucocyte adhesion and extravasation. ECM synthesis, the expression of adhesion molecules and cytokines, the increase of EC permeability, and the increase of leukocyte adhesion and extravasation contribute to lung injury as a result of ICAM1 clustering [41]. The ECM stiffness inhibitors, such as curcumin, can be used in the treatment of age-related diseases [42,43].

### 2.6. Substrate Stiffness Can Affect Both Extracellular Vesicles Secretion and Uptake

Microvesicles play important role in the progression of various diseases, including COVID-19. Microvesicles are observed at elevated concentrations in diseased tissues [44]. It has been reported that matrix stiffness regulates microvesicle-induced fibroblast activation. It has been observed that microvesicles released by malignant breast cancer cells lead to an increase in fibroblast spreading and to α-smooth muscle actin expression [45]. Lower nanoparticle uptake efficiency by breast cancer cells on the soft matrix was observed compared with nanoparticle uptake on stiffer matrices [46,47]. So, it may be expected that virus uptake will be higher in cells localized in the stiffer tissues of elderly patients when compared with virus uptake by cells localized in softer tissues of healthy young people. However, it was reported that extracellular vesicles uptake by cells is higher on soft surfaces compared with extracellular vesicles uptake when cells are cultured on stiff matrices [48]. It has been concluded that substrate stiffness can affect both extracellular vesicles secretion and uptake. Exosome-based strategies for the treatment of COVID-19 virus infection may include the inhibition of exosome uptake [49]. It has been reported that human umbilical cord mesenchymal stem cell (UMSC)-derived exosomes prevent aging-induced cardiac dysfunction by releasing novel lncRNA metastasis-associated lung adenocarcinoma transcript 1 (MALAT1), which inhibits the NF-κB/TNF-α signaling pathway [50]. It can be expected that exosomes may be investigated for possible application also in anti-TNF-α therapies as the inhibitors of the NF-κB/TNF-α signaling pathway. So, the maintaining tissue integrity plays very important role in reducing poor outcomes in COVID-19.

### 2.7. Glycocalyx Degradation Depends on Tisue Stiffness

Weinbaum and co-authors recently concluded that arterial stiffness is associated with glycocalyx degradation, endothelial dysfunction, and vascular disease [51]. Increased arterial stiffness has been linked with impaired endothelial glycocalyx. ACE2 plays important role linking SARS-CoV-2 infection, cardiovascular diseases, and immune response. In COVID-19, endothelial glycocalyx shields the interaction of the SARS-CoV-2 spike protein with ACE2 receptors, thus preventing the development of the disease. The probability of COVID-19 infection depends on the glycocalyx thickness. A decrease in glycocalyx layer thickness exposes ACE2 receptors and promotes their interaction with the viral Spike protein [52]. A decrease in the binding energy of water molecules and the transformation of tightly bound water into loosely bound, and free water [3] results in the increased binding of spike S-1 protein to ACE2 receptors with increased COVID-19 infection. So, it would be promising to investigate the relationship between the content of tightly bound water and viral binding of spike protein. Molecular dynamics simulations demonstrated the higher affinity of SARS-CoV-2 to ACE2 as compared to SARS-CoV and rigidity of spike protein plays important role in the interaction of host cell and virus [53]. It has been revealed that kobophenol A inhibits the binding of the host ACE2 receptor with the spike RBD domain of SARS-CoV-2 for blocking COVID-19 [54]. Metformin is an antidiabetic drug with still not well recognized underlying mechanisms for health beneficial effects. Targosz-Korecka et al. found using atomic force microscopy (AFM) that metformin decreases arterial stiffness [55,56]. It has been suggested that decrease in matrix stiffness may be beneficial in the prevention and treatment of COVID-19 [3]. Metformin can induce glycocalyx restoration, decreasing stiffness and preventing disease development. The recovery of endothelial glycocalyx studied using nanoindentation experiments and confocal microscopy imaging was correlated with the decreased expression of cell adhesion molecules E-selectin and ICAM-1. ICAM-1 expression can be associated with metastases and poor prognosis in various cancers, such as melanoma, breast, lung, and oral cancer. The potential to repurpose metformin for application in the treatment of COVID-19 has been suggested [57]. Reduced endothelial glycocalyx thickness was associated with increased pulse wave velocity [58]. COVID-19, which is closely related with suggested increased tissue stiffness, results in glycocalyx damage [59]. Glycocalyx can play important role in the prevention of thrombotic complications [60]. This hydrated gel-like layer at the blood–endothelium interface regulates vascular permeability [58]. Glycocalyx deterioration was associated with aging and cardiovascular diseases [61], diabetes [62], and chronic kidney disease [63]. Glycocalyx deterioration, similarly to the increase of tissue stiffness, results in increased tissue permeability. Endothelial glycocalyx destruction is related with increased vascular permeability and impaired anti-coagulation as a result of endothelium damage. Anti-oxidants, doxycycline, etanercept, hydrocortisone, and antithrombin III can be used to prevent or treat endothelial glycocalyx damage [64,65]. Etanercept also decreases arterial stiffness in rheumatoid arthritis patients [66]. The inhibition of IL-6 activity by tocilizumab decreases arterial stiffness and improves endothelial glycocalyx in patients with rheumatoid arthritis [67]. Sarilumab and siltuximab, as well as tocilizumab, are Food and Drug Administration (FDA)-approved IL-6 inhibitors that can be used by patients with COVID-19 [68]. Anakinra is a recombinant IL-1 receptor antagonist that has been suggested in the treatment of the cytokine storm associated with SARS-CoV-2 [68]. The role of albumin in the preservation of endothelial glycocalyx integrity has been reviewed [69]. Sulodexide, a mixture of low-molecular weight heparin (80%) and dermatan sulfate (20%), an oral preparation, has been shown to restore endothelial glycocalyx after 2 months of therapy [70]. Zhang et al. described the use of liposomal nanocarriers comprising preassembled glycocalyx to restore damaged endothelial glycocalyx [71]. It is evident that glycocalyx restoration is important and promising emerging area of research. Constantinescu et al. [72] demonstrated that oxidized LDL (Ox-LDL) degrades the negatively charged endothelial glycocalyx and stimulating leukocyte-endothelial cell adhesion, but supplementation with sulfated polysaccharides attenuates Ox-LDL-induced leukocyte-endothelial cell adhesion. Shimoda and co-authors reported that polythenols ferulic acid, quercetin, and kaempferol decreased glycocalyx damage and enhanced the expression of hyaluronan binding protein 2 and hyaluronan synthase 2 in adiponectin-treated dermal fibroblasts, increasing hyaluronan production [73].

It is also known that associations exist between higher flavonoid consumption and a lower arterial stiffness [74]. Researchers at Department of Nutrition and Integrative Physiology in Florida State University reported that microcirculatory and glycocalyx properties are lowered by a high-salt diet but augmented by the consumption of diets high in fat, sugar, and salt (a Western diet) in genetically heterogeneous mice [75]. The dimension of the endothelial glycocalyx measured by confocal microscopy in Sprague Dawley male rats fed a 12 week high-cholesterol diet decreased significantly compared with the control [76]. So, diets are able to regulate the stiffness and related health risks. Glycocalyx regeneration therapy, including anticoagulants, such as heparin, nutraceuticals, and pharmaceuticals, has been discussed [77].

Cholesterol content increases in the membranes of cells with aging. Oxidized low-density lipoprotein (LDL) cholesterol is vasoconstrictor and thrombogenic. The synthesis and release of nitric oxide is inhibited by LDL cholesterol [78]. Statins can decrease the content of cholesterol in the membranes of cells and the reduced content of cholesterol can prevent viruses from entering cells [79]. Membrane cholesterol depletion results in decrease of stiffness, adhesion, and cytoskeletal disorganization. Cholesterol enrichment leads to an increase in stiffness, adhesion, and the cytoskeletal organization of cells [80,81]. The depletion of membrane cholesterol inhibits SARS-CoV-2 and other coronaviruses [82].

Chitosan and chitosan oligosaccharides inhibit pro-inflammatory cytokines, adhesion molecules and nuclear factor NF-κB activation, and the translocation of NF-κB from cytoplasm to nucleus which has been associated with the development of age-related diseases [83,84,85,86,87,88]. Thus, it can be expected that the combination of chitosan, chitosan oligosaccharides, and polyphenols can prevent the development of severe COVID-19 due to the inhibition of NF-κB and TNF-α, which lead to the increase of tissue stiffness and permeability.

### 2.8. Anti-TNF-α Therapy Decreases ECM Stiffness and Can Be Used in the Treatment of Age-Related Diseases

Researchers at the University of Cambridge, Addenbrooke’s Hospital, United Kingdom, demonstrated that arterial stiffness measured by pulse-wave velocity and related diseases can be treated and reduced using anti-TNF-α therapy [89]. The pro-thrombotic effects of TNF-α have been discussed [90], and TNF-α can lead to diabetes, cardiovascular disease, and to severe COVID-19 [3], as seen in Figure 1 and Figure 2. Repurposing anti-TNF as a therapy for the treatment of COVID-19 has been suggested in a number of recent publications [91,92,93,94,95]. It is important to understand that anti-TNF-α therapies in fact are also destiffening therapies because TNF-α increases tissue stiffness, decreases tissue integrity, and leads to cytokine storm. The association of TNF-α with arterial stiffness has been reported [95]. Substrate stiffness regulates the expression of inflammatory genes via actomyosin contractions, which triggers NF-κB activation [96]. The inhibition of NF-κB pathway also must be investigated in more detail as a potential therapy of the severe form of COVID-19.

It has been also reported in a number of publications that various polyphenol compounds inhibit TNF-α in vitro [97,98,99,100,101,102,103,104]. ACE2 hydrolyzes vasoconstricting angiotensin II generating vasodilating angiotensin (1–7). A computational study demonstrated that flavonoids curcumin and catechin establish hydrogen bonds with the ACE2 of the human cell membrane and interact with corona viral S protein reducing viral entry. Polyphenols decrease tissue stiffening, so it could be suggested that polyphenols indirectly reduce viral entry [103]. Recently, it has been demonstrated that blueberry-derived exosome-like nanoparticles counter the response to TNF-α-induced change on gene expression in EA.hy926 cells [104]. It has been recently reported that a number of circulating endothelial progenitor cells are increased in COVID-19 patients [105]. Endothelial progenitor cells may be protected by resveratrol from TNF-α-induced inflammatory damage [106]. So, the application of anti-TNF-α therapy for the protection of endothelial progenitor cells can be justified. Endothelial progenitor cells play an integral role in the cellular repair but their number in circulation and function decrease with aging and increase vascular disease risk [107]. Obesity suppresses the circulating level and function of endothelial progenitor cells [108]. Patients with diabetes exhibit the reduced number and function of endothelial progenitor cells [109]. Endothelial progenitor cells bind and inhibit platelet activation, aggregation, adhesion to collagen via the upregulation of cyclooxygenase-2 and the secretion of prostacyclin, thus preventing thrombus formation [110]. Inflammation and reactive oxygen species decrease the number of circulating endothelial progenitor cells [111]. Age-related inflammation and enhanced oxidative stress are associated with tissue stiffness [3]. Therefore, it can be expected that increase in tissue stiffness leads to the decrease in the number and activity of endothelial progenitor cells.

Statins, angiotensin II receptor 1 blockers, and ACE2 exert beneficial effects on endothelial progenitor cells [112]. At the same time, statins can also be used to decrease tissue stiffness [113], as well as angiotensin II receptor blockers and ACE2 [114]. So, the same drugs improve both tissue stiffness and endothelial progenitor cell function.

Aspirin decreases arterial stiffness [115]. The enhanced platelet turnover and reactivity may be associated with disease severity in COVID-19 patients. Aspirin decreases platelet turnover and prevents thrombus formation [116]. Aspirin mitigates severe COVID-19 complications [117]. Thus, it can be expected that, coupled with other effective drugs, aspirin could reduce death rate in critically ill COVID-19 patients. COVID-19-associated coagulopathy has been recently reviewed [118].

It has been suggested that SARS-CoV-2 and its Spike protein directly stimulated the release of coagulation and inflammatory factors and the formation of leukocyte-platelet aggregates [119].

However, it was previously reported that platelets sense microenvironmental mechanical properties [20] and substrate stiffness during clot formation mediates adhesion, spreading, and activation. The effect of arterial stiffness on platelet activation has also been reported in a number of publications [120,121,122].

### 2.9. Oxidative Stress and Matrix Stiffness

Cardiovascular diseases are associated with increased oxidative stress [123], and cardiovascular diseases are associated with arterial stiffness [124]. Oxidative stress is also associated with changes in platelet function [125] and platelet adhesion depends on substrate stiffness [20]. Polyphenols are antioxidants [126], and at the same time, polyphenols decrease arterial stiffness [127]. Dietary polyphenols (i.e., flavones, isoflavones, flavonols, catechins, and phenolic acids) reduce both oxidative stress and thrombotic risk, preventing oxidative stress-induced platelet activation [126].

This means that the evident relationship between oxidative stress and matrix stiffness must be expected. Reactive oxygen species (ROS) promote arterial stiffening [128] and changes in substrate stiffness change ROS generation [129]. For example, higher liver stiffness facilitates hepatocellular carcinoma (HCC) development and progression. An increase in ROS production in HCC cells under higher stiffness stimulation was reported [130].

P-selectin may be proposed as a marker of endothelial dysfunction and increased arterial stiffness in hypercholesterolemic patients. Oxidative processes leading to endothelial dysfunction and platelet activation result in increased soluble P-selectin levels [131]. P-selectin is partially responsible for the adhesion of leukocytes and platelets to the endothelium [132]. It has been also reported that serum sP-Selectin level was 1.7 ng/mL in the control group (1–3.78), 6.24 ng/mL (5.14–7.23) in the mild-to-moderate pneumonia group, and 6.72 ng/mL (5.36–8.03) in the severe pneumonia group in COVID-19 disease [133]. Thus, increased arterial stiffness and increased sP-Selectin level was observed both in cardiovascular diseases and in COVID-19. sP-Selectin can be considered as an important molecule in the development of arterial stiffness [134]. However, at present, the role of matrix stiffness is often underestimated and even ignored during the consideration of biological processes. In many cases, stiffening plays a decisive role but is mistakenly considered as a secondary process dependent on other primary processes.

A correlation has been found between arterial stiffness determined by pulse wave velocity and platelet activation [28].

It has been reported that oxidative stress by Nox2 activation is associated with severe disease and thrombotic events in COVID-19 patients [135].

Reciprocal dependence between TNFα- and NOX2 has been reported [136], evidencing the interdependence between oxidative stress and matrix stiffness. Both NOX2 ROS and TNFα played an important role in the observed aortic dysfunction. The authors concluded that the release of TNFα causes perivascular adipose tissue ROS production through the activation of the NOX2-dependent pathway, activates aortic ROS production, and mediates aortic stiffness. The neutralization of TNFα and/or the inhibition of NOX2 prevents the impairment of aortic function by perivascular adipose tissue. Eliminating Nox2 ROS resulted in reduced tissue stiffness.

High levels of 8-iso-PGF2α in COVID-19 have been reported [137], indicating increased oxidative stress. High oxidative stress can be predicted in COVID-19 patients because increased arterial stiffness was reported in COVID-19 patients [22].

Oxidative stress activity based on the level of 8-iso-PgF2α; pro-inflammatory activity based on tumor necrosis factor-α, its type I soluble receptor; and C-reactive protein levels and the effects of antihypertensive treatment on systemic inflammation, oxidative stress, and pro-inflammatory cytokine levels in 186 hypertensive patients during a 2-month course of treatment have been reported with aim of identifying the possible role of oxidative stress in hypertension. It was found that the grade of hypertension influences the increase in the serum levels of TNF-α and 8-iso-PgF2α, and the duration of hypertension affects the activity of the TNF- α with no effect on the levels of 8-iso-PgF2α [138]. The similar dependence of TNF-α and 8-iso-PgF2α on the grade of hypertension demonstrated the relationship between matrix stiffness and oxidative stress.

It has been stated that vitamins improve arterial stiffness due to destiffening [139]. So, the conclusion can be reached that various vitamins can decrease the severity of COVID-19. The positive effect of various vitamins on COVID-19 has also been confirmed in a number of publications [3,11,140,141,142,143,144]. It has been concluded that vitamins play an important role in the regulation of the immune response. However, it could be suggested that destiffening also plays an essential role in this effect.

Physical activity decreases the risk for severe COVID-19 outcomes and alleviates post-acute COVID-19 syndrome [121]. Such conclusions could be foreseen because it had previously been reported that habitual physical activity is associated with lower arterial stiffness in older adults [145,146].

## 3. Conclusions and Future Perspectives

Extracellular matrix stiffening, linked with inflammation and oxidative stress, is a driving factor of various age-related diseases, including cardiovascular pathologies and very likely COVID-19. Efforts to find effective drugs that are able to fight severe COVID-19 and post-COVID-19 complications are expected to remain to be a hot topic in the near future.

This review was limited to the articles containing information about the matrix stiffness effect on cellular processes and age-related diseases. The increased tissue stiffness and related tissue integrity disruption and increased membrane cholesterol content in severe COVID-19 patients may be associated with increased virus entry into host cells and increased disease severity. Cholesterol content in cell membrane increases with aging and with ECM stiffening. Statins for cholesterol depletion and arterial stiffness reduction and anti-TNF-α therapies for cytokine storm prevention and arterial stiffness reduction in COVID-19 could be suggested for more detailed investigation. Local stiffness growth at the sites of rigid virus particle penetration into vascular walls can trigger mechanosensitive platelet activation and the initiation of thrombus formation in critically ill COVID-19 patients.

Glycocalyx degradation in severe COVID-19 patients has been related with increased arterial stiffness. The inhibition of NF-κB and TNF-α can improve tissue integrity and biomechanical properties. Heparin and potentially sulfated chitosans can be used in the future for the treatment of stiffness-induced thrombotic complications in COVID-19. Aspirin prevents thrombus formation by decreasing platelet turnover and mitigates severe COVID-19 complications. Various diets can regulate stiffness and related health risks.

The molecular mechanism is still poorly understood. Very different experimental results from various areas of research presented in this review taken together provide evidence that ECM stiffness plays an important and often decisive role in the cellular processes responsible for age-related and age-dependent diseases, including COVID-19. The association of arterial stiffness with inflammation and oxidative stress can give new insights into the mechanisms of thrombus formation. Prospective, randomized, and well-designed studies would be beneficial to develop a synergistic combination of natural and synthetic molecules for the potential treatment of COVID-19 and post-COVID-19 patients, provided that proper evidence of their efficacy and safety is established. Therefore, larger prospective studies on the assessment of arterial stiffness and tissue integrity should be performed in order to gain more breakthrough knowledge related to the prevention and treatment of severe COVID-19 infection.

## Figures and Tables

**Figure 1 ijms-24-01187-f001:**
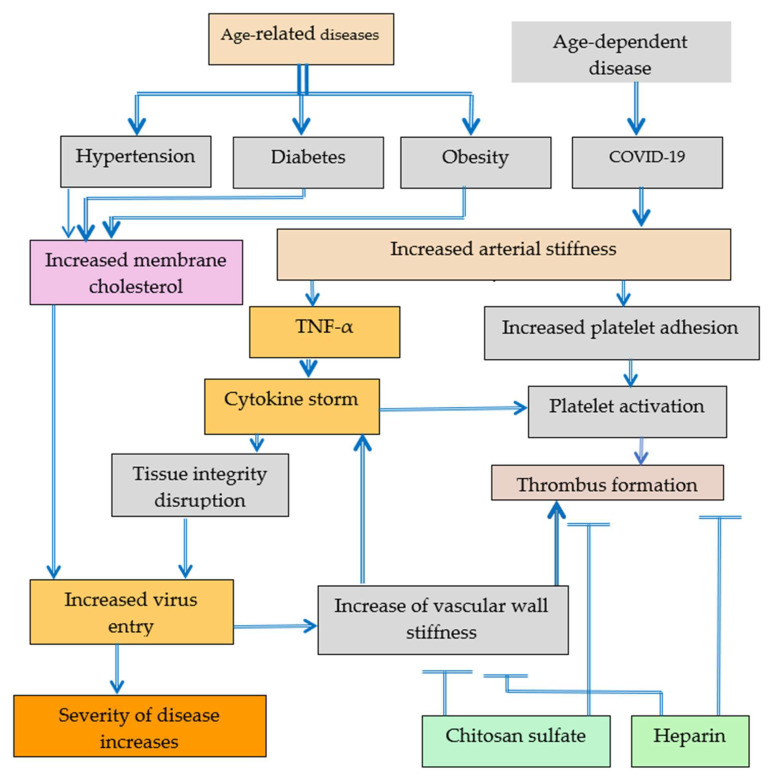
Platelet activation leading to thrombus formation can be caused by increased arterial stiffness in COVID-19 patients or by cytokine storm also induced by increased arterial stiffness. Patients with hypertension, diabetes, and obesity are more often severely ill with COVID-19 probably because of higher old age-related tissue stiffness in various organs. Partly the increased tissue stiffness can be explained by increased aging cholesterol content in cell membranes and the related increase of virus entry in cells.

**Figure 2 ijms-24-01187-f002:**
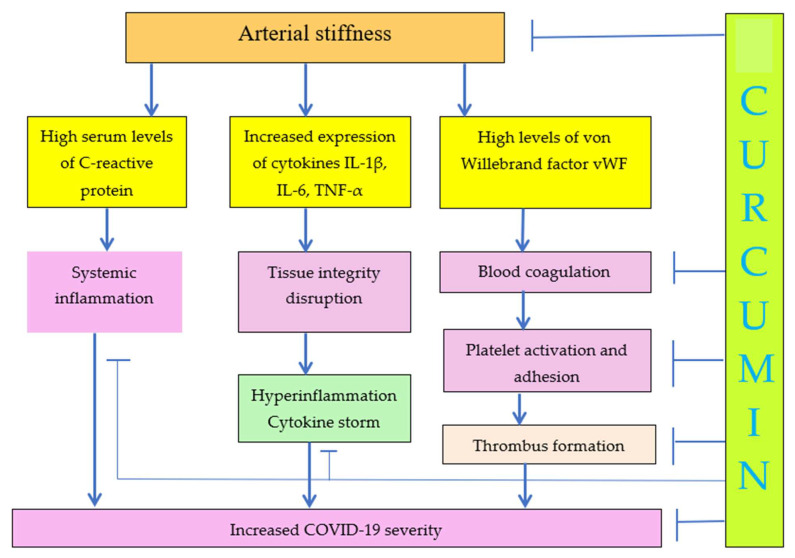
Increase of ECM stiffness results in increased COVID-19 severity. Curcumin decreases arterial stiffness and COVID-19 severity due to decrease of inflammation, cytokine storm, blood coagulation, and platelet activation.

**Figure 3 ijms-24-01187-f003:**
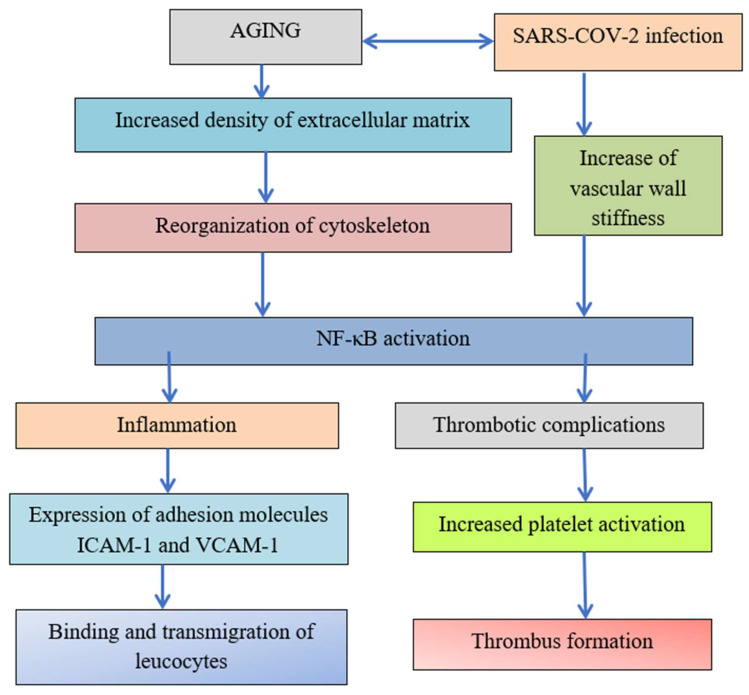
The transcription factor NF-κB has links to thrombotic processes and inflammation.

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
