# Peer review of "Severe COVID-19—A Review of Suggested Mechanisms Based on the Role of Extracellular Matrix Stiffness"

_ijms, 2023, doi:10.3390/ijms24021187_

Round 1

Reviewer 1 Report

Dear Editor and author(s),

thank you for the opportunity to review this very interesting paper. It provides an overview of potential mechanisms, and took large effort to be put on paper in an organized manner. I have several but minor suggestions:

1) when considering adhesion molecules and their expression in patients with COVID in the introduction section, it is unclear whether expression refers to specific tissues, white blood cells, protein expression in plasma or something else.

2) Platelets play important role not only in thrombosis and hemostasis processes, but also in lung development and regeneration. Thus they gain attention during COVID pandemic. I suggest to include comment in the introduction section about large retrospective registry based study (Blood Cancer J. 2021 Nov 29;11(11):189. doi: 10.1038/s41408-021-00585-2) reporting that both high MPV and low platelets are independently associated with inferior survival in mostly severe/critical hospitalized COVID-19 patients but also higher platelets are associated with higher rates of venous thromboembolism, whereas higher MPV is associated with higher rates of arterial thrombotic events. 

3) In the conclusion section, paper is missing critical disclaimer regarding no high quality evidence for large majority of mentioned compounds, drugs and drug classes to actually improve outcomes of severe/critical COVID patients. Thus paper should explicitelly state that prospective, hopefully randomized, well designed studies would be needed to establish place of these molecules in potential treatment of COVID patients. This review is primarily focused on providing overview of very interesting concept of active participation of matrix stiffness in pathophysiology of COVID-19 and not on promoting therapy with any of these compunds or drugs until proper evidence of their efficacy and safety is established.

Author Response

Thank you very much for appropriate and helpful comments.

1) when considering adhesion molecules and their expression in patients with COVID in the introduction section, it is unclear whether expression refers to specific tissues, white blood cells, protein expression in plasma or something else.

Expression of adhesion molecules by endothelial cells has been added in the Introduction section.

2) Platelets play important role not only in thrombosis and hemostasis processes, but also in lung development and regeneration. Thus they gain attention during COVID pandemic. I suggest to include comment in the introduction section about large retrospective registry based study (Blood Cancer J. 2021 Nov 29;11(11):189. doi: 10.1038/s41408-021-00585-2) reporting that both high MPV and low platelets are independently associated with inferior survival in mostly severe/critical hospitalized COVID-19 patients but also higher platelets are associated with higher rates of venous thromboembolism, whereas higher MPV is associated with higher rates of arterial thrombotic events.

Thank you. The above citation and appropriate text have been added to Introduction. 

3) In the conclusion section, paper is missing critical disclaimer regarding no high quality evidence for large majority of mentioned compounds, drugs and drug classes to actually improve outcomes of severe/critical COVID patients. Thus paper should explicitelly state that prospective, hopefully randomized, well designed studies would be needed to establish place of these molecules in potential treatment of COVID patients. This review is primarily focused on providing overview of very interesting concept of active participation of matrix stiffness in pathophysiology of COVID-19 and not on promoting therapy with any of these compunds or drugs until proper evidence of their efficacy and safety is established.

I agree completely with you. The suggested text has been added to Conclusion.

Reviewer 2 Report

The article in its present form is extremely hard to comprehend. Though there is a good amount of relevant information, however, the manner in which the information has been conveyed looks vague. It would require extensive editing to make it understandable. 

Author Response

The article in its present form is extremely hard to comprehend. Though there is a good amount of relevant information, however, the manner in which the information has been conveyed looks vague. It would require extensive editing to make it understandable. 

This review is rather critical than comprehensive. So early-career researchers and newcomers due to their limited knowledge must apply some effort and beneficially improve their knowledge and expertise in order to adequately comprehend the current yet developing state-of-the-art described in the review. The target readers to a great extent are mostly experienced researchers that currently participate in elaboration of the wright concept in the area of emerging knowledge and are able to make valuable inputs in potential international projects in the nearest future by selection of suitable collaboration partners based on the provided review. The provided by you non justified and evidently unfair scores demonstrate that your knowledge and expertise in the field of review is limited, if any, that you are angry (why ???), and it is evidently hard for you to provide reasonable suggestions, but fortunately other 4(!) reviewers have complementary expertise and recommended some positive and valuable additions in the review text that have been taken into account thankfully and can be of interest for experienced researchers that will be able to develop in the nearest future this emerging field with high potential biomedical input.

Editing of manuscript has been made in line with specific comments of 4 reviewers.

Reviewer 3 Report

Remarks:

In this review the author, Kerch has discussed on matrix stiffness effect on cellular processes and age-related diseases in relation to severity of COVID-19 infection in patients. There the author has reviewed the relationship between dehydration, TNF-α activation, NF-κB links to thrombotic processes and inflammation, RHOA activity, extracellular vesicles secretion and uptake, glycocalyx degradation, anti-TNF-α therapy and oxidative stress, and the tissue stiffness which could affect the severity of COVID-19 infection. Thus, Kerch suggests that several drugs, including membrane cholesterol lowering drugs, sulfated chitosan derivatives, anti-TNF-α therapies and also, inhibition of the nuclear factor kappa-light-chain-enhancer of activated B cells pathway can be considered as a therapeutic target in the treatment of severe COVID patients.

Comments:

1.              Although the author has reviewed the COVID-19 virus entry enhancement in relation to tissue stiffness, it is not clear how the tissue stiffness will contribute to the virus entry into cells via the ACE2 receptors (the relevant receptor for COVID-19 virus entry). Therefore, author need to add an additional section to explain it.

2.              Figure 1 – The figure is not completed and need to be revised.

3.              There are inconsistencies in using abbreviations (eg: line no 10) and spellings (eg: line no 12). Author should carefully read and revise the manuscript accordingly.

Thank you.

Author Response

  1. Although the author has reviewed the COVID-19 virus entry enhancement in relation to tissue stiffness, it is not clear how the tissue stiffness will contribute to the virus entry into cells via the ACE2 receptors (the relevant receptor for COVID-19 virus entry). Therefore, author need to add an additional section to explain it.

Thank you for your comment. It has been added in part 2.8. that ACE2 hydrolyzes vasoconstricting angiotensin II generating vasodilating angiotensin (1-7). A computational study demonstrated that flavonoids curcumin and catechin establish hydrogen bonds with ACE2 of human cell membrane and interact with corona viral S protein reducing viral entry. Polyphenols decrease tissue stiffening, so it may be suggested that indirectly polyphenols reduce viral entry.

It has been also mentioned at the end of section 2.2. with citation references 25-27 the possible indirect role of stiffness and related cytokine storm on ACE2 receptors. 

  1. Figure 1 – The figure is not completed and need to be revised.

 Figure 1 is based on conclusions reported in cited publications and demonstrates relationships that have been suggested in reviewed publications by different authors. Since the  research in field of COVID-19 is still in progress and not yet completed and positive problem solution is expected only in future this figure is not final and will be modified with the progress in research results. For example, the novel factors that increase vascular wall stiffness and their inhibitors as potential more effective drugs in treatment of age related diseases and COVID-19 must be clearly identified and described in detail in the nearest future in order to be added to Figure 1 and to the text of review.

  1. There are inconsistencies in using abbreviations (eg: line no 10) and spellings (eg: line no 12). Author should carefully read and revise the manuscript accordingly.

The text has been carefully checked and the found errors have been corrected

Reviewer 4 Report

This review aims to analyze the influence of micro-environmental stiffness on various mechanisms of cellular processes involved in COVID-19 disease. 

The abstract section should be improved. I suggest structuring it as a review (Background, aims, different sections, and conclusion). Please, revisit it.

In the introduction section, the aims of the review should be better described. Moreover, the authors should clarify the methodology of the review (for example, the authors should clarify the inclusion and exclusion criteria). This point is crucial: please, clarify the methodology.

In section 2.3 it could be useful to introduce the hyperexpression of FVIII in COVID-19 subjects. In this way, I suggest several seminal references: DOI:10.3390/diagnostics10080575; DOI:10.1182/blood.2020007335; DOI:10.3389/fimmu.2021.649122. Please, analyze this missed point.

In section 2.9, I suggest remarking on the importance of dietary supplementation. In this way, I suggest several seminal references: DOI; 10.1080/14787210.2022.2125867; DOI: 10.3390/nu13030976; DOI: 10.1177/15598276221140864. In the same section, the author missed analyzing the role of physical activity in order to prevent severe ill in COVID-19 patients.

Minor points:

- please, insert the figure in the proximity of the text citation.

- "FIGURE 3" is reported in capital letters.

Author Response

The abstract section should be improved. I suggest structuring it as a review (Background, aims, different sections, and conclusion). Please, revisit it.

The structured abstract is not in line with the Guide for Authors in this Journal but I made changes to improve the Abstract. 

In the introduction section, the aims of the review should be better described. Moreover, the authors should clarify the methodology of the review (for example, the authors should clarify the inclusion and exclusion criteria). This point is crucial: please, clarify the methodology.

It has been added in Introduction that the effect of mechanical forces and extracellular matrix stiffness on biological cellular processes was the criterium for published data inclusion in this review. The publications related with direct and indirect destiffening strategies and destiffening agents are of great interest. 

In section 2.3 it could be useful to introduce the hyperexpression of FVIII in COVID-19 subjects. In this way, I suggest several seminal references: DOI:10.3390/diagnostics10080575; DOI:10.1182/blood.2020007335; DOI:10.3389/fimmu.2021.649122. Please, analyze this missed point.

Thank you. The additions in section 2..3 have been made according to your recommendation.

In section 2.9, I suggest remarking on the importance of dietary supplementation. In this way, I suggest several seminal references: DOI; 10.1080/14787210.2022.2125867; DOI: 10.3390/nu13030976; DOI: 10.1177/15598276221140864. In the same section, the author missed analyzing the role of physical activity in order to prevent severe ill in COVID-19 patients

Thank you. The additions have been made in section 2.9 according to your recommendation.

Reviewer 5 Report

Here the authors investigate why covid -19 lead to thrombosis. A question which there is no judging answer for. This review articles bring together a good base of all clues existent for solve this enigma.

I propose to the author to consider these comments to make it clearer for reader:

Major comments

Working on their charts: Figure 1: For make it better understanding and more informative, I suggest to author to add to their chart the cytokine or at least the name of pathway triggered in each square. I propose also to explain better their figure in legend. This figure could be a pipeline to understand or categorized the thrombosis symptoms detected in patients. Other point in this figure, I am not sure that platelet adhesion are upstream of platelet activation. I can understand that authors may consider the adhesion of platelets to endothelial cells as the result of stiffness, but I propose to explain better.

Same points for figure 2, based on the chart, increase in level of vWF result in platelet activation, but it needs to be more accurate and show the different step of coagulation and platelet activation which are 2 different parts of thrombosis.

Overall, I think that thrombosis paly a crucial role in this article and the author has right to explain the mechanism and basis of platelet activation and coagulation (two major events) in a paragraph.

Minor comment

Line 42,44 need editing.

Author Response

Here the authors investigate why covid -19 lead to thrombosis. A question which there is no judging answer for. This review articles bring together a good base of all clues existent for solve this enigma.

I propose to the author to consider these comments to make it clearer for reader:

Thank you very much for your comments..

Major comments

Working on their charts: Figure 1: For make it better understanding and more informative, I suggest to author to add to their chart the cytokine or at least the name of pathway triggered in each square. I propose also to explain better their figure in legend. This figure could be a pipeline to understand or categorized the thrombosis symptoms detected in patients. Other point in this figure, I am not sure that platelet adhesion are upstream of platelet activation. I can understand that authors may consider the adhesion of platelets to endothelial cells as the result of stiffness, but I propose to explain better.

Figure 1 is based on conclusions reported in cited publications and demonstrates relationships that have been suggested in reviewed publications of different authors. Since the  research in field of COVID-19 is still in progress and not yet completed and positive problem solution is expected only in future this figure is not final and will be modified with the progress in research results. I provided more detailed explanations according to your recommendations in the legend.

Figure 1. Patients with hypertension, diabetes, and obesity more often are ill by severe COVID-19 probably because of higher old age-related tissue stiffness in various organs. Partly the increased tissue stiffness can be explained by increased with aging cholesterol content in cell membranes and related increase of virus entry in cells.

Same points for figure 2, based on the chart, increase in level of vWF result in platelet activation, but it needs to be more accurate and show the different step of coagulation and platelet activation which are 2 different parts of thrombosis.

The modifications have been provided. The legend has been extended and the effect of curcumin on decreases of arterial stiffness and COVID-19 severity due to decrease of inflammation, cytokine storm, blood coagulation and platelet activation recently reported in clinical trial in 2022.

Overall, I think that thrombosis paly a crucial role in this article and the author has right to explain the mechanism and basis of platelet activation and coagulation (two major events) in a paragraph.

The text has been added to the legend of Figure 2 and Figure has been modified with the aim to demonstrate that polyphenol curcumin due to decrease of arterial stiffness also inhibits inflammation, cytokine storm, coagulation, platelet activation and thrombus formation 

Minor comment

Line 42,44 need editing.

All the manuscript text has been edited and found errors have been corrected.

Round 2

Reviewer 2 Report

I have nothing more to comment on this manuscript.

Reviewer 4 Report

Following the reviewers' suggestions, the authors have improved their manuscript.